# Selective Catalysis by Complexes Including Ni and Redox-Inactive Alkali Metals (Li, Na, or K) in Oxidation Processes: The Role of Hydrogen Bonds and Supramolecular Structures

**DOI:** 10.3390/ijms26031166

**Published:** 2025-01-29

**Authors:** Ludmila I. Matienko, Elena M. Mil, Anastasia A. Albantova, Alexander N. Goloshchapov

**Affiliations:** N.M. Emanuel Institution of Biochemical Physics Russian Academy of Science, 4 Kosygin Str., 119334 Moscow, Russia

**Keywords:** triple binuclear complexes Ni(acac)_2_∙MSt∙PhOH (M = Li, Na, or K), H bonds, catalysis, oxidation, nanostructures, NiARD models

## Abstract

It is known that the presence of redox-inactive metals in the active center of an enzyme has a significant effect on its activity. In this regard and for other reasons, the effect of redox-inactive metals on redox processes, such as electron transfer, oxygen and hydrogen atom transfer, as well as the breaking and formation of O–O bonds in reactions catalyzed by transition metals, has been widely studied. Many questions about the role of redox-inactive metals in the mechanisms of these reactions remain open. In this paper, the mechanism of catalysis by bi- and triple hetero-binuclear heteroligand complexes including Ni and redox-inactive alkali metals ((A) {Ni(acac)_2_∙L^2^} and (B) {Ni(acac)_2_∙L^2^∙PhOH} (L^2^ = MSt (M = Li, Na, or K)) in the process of the selective oxidation of ethylbenzene by molecular oxygen into α-phenyl ethyl hydroperoxide is considered. The activity of A and B complexes towards O_2_, ROOH, and RO_2_^•^ radicals was studied. Based on kinetic data, we suggest that the high catalytic efficiency of B triple complexes in oxidation processes may be associated with the role of outer-sphere regulatory interactions, with the formation of stable supramolecular structures due to intermolecular H bonds. This assumption was confirmed using the AFM method. Prospects for studying catalysis by complexes ({Ni(acac)_2_∙L^2^} and {Ni(acac)_2_∙L^2^∙PhOH}) that are models of NiARD (Ni-Acyreductone dioxygenase) are discussed.

## 1. Introduction

It is known that compounds of redox-inactive metals in combination with redox-active transition-metal complexes are used to promote a variety of redox reactions [1]. The effects of metal ions on electron-transfer reactions from electron donors (Ds) to electron acceptors (As) is typified in metalloproteins such as copper and zinc superoxide dismutase, in which both metal ions have been proposed to be functionally active [1,2]. Cu/Zn-dependent superoxide dismutase (Cu/Zn-SOD) catalyzes the disproportionation of superoxides (O_2_^•–^) to H_2_O_2_ and O_2_ at an active site containing a redox-inactive Zn^2+^ [2]. Redox-inactive and -active metal ions are essential cofactors in modulating the redox reactivity of metal–oxygen complexes and metalloenzymes, such as the manganese (V)–oxo intermediate in the oxygen-evolving complex. The Ca^2^⁺ ions in the Mn_4_CaO_5_ cluster in Photosystem II catalyzes the oxidation of H_2_O to O_2_ [3,4].

In this regard, the influence of redox-inactive metals on various transition metal-mediated redox processes has been widely studied [3].

Lewis acidic metal centers can profoundly affect redox processes even without the intrinsic ability to directly participate in electron transfer. Upon coordination with redox-active metal complexes, these Lewis acids impact a range of properties, including the reduction potential of transition metal centers, the electrophilicity and nucleophilicity of metal–oxo, –hydroxo, –peroxo, and –superoxo species, and the steric profile of these motifs. These effects can contribute to the enhancement of the activity of otherwise unreactive species, prompt shifts in the mechanism of certain transformations, and help direct product selectivity in various chemical reactions [3]. For example, in the absence of redox-inactive metal ions, the manganese(II) catalyst is ineffective in epoxidizing cyclooctene. Only 9.9% cyclooctene conversion and a 4.1% epoxide yield were observed. EPR studies of the manganese(II) catalyst in the presence of an oxidant clearly indicate the formation of a mixed-valent di-μ–oxo-bridged diamond core, Mn^III^-(μ-O)_2_-Mn^IV^. However, the addition of 2 equivalents of Al^III^ to the manganese(II) catalyst sharply improves the epoxidation, providing up to 97.8% conversion with a 91.4% yield of epoxide. The presence of a Lewis acid like Al^III^ causes the dissociation of this diamond Mn^III^-(μ-O)_2_-Mn^IV^ core to form a monomeric manganese(iv) species which is responsible for improved epoxidation efficiency [5].

The adducts of salts of redox-inactive metals with β-diketonates and N,N′-ethylene bis (salycylideniminates) of Co^II^, Ni^II^, and Cu^II^ are known [6]. For example, from Schiff bases, H_2_L^1^[N,N′-bis(3-methoxysalicylidene)propane-1,3-diamine] and H_2_L^2^[N,N′-bis(3-methoxysalicylidene)-2,2-dimethylpropane-1,3-diamine] have been prepared as hetero-dinuclear nickel(II)/zinc(II) complexes, [(DMSO)_2_NiL^1^Zn(NCS)_2_] and [(DMSO)_2_NiL^2^Zn(NCS)_2_]. Single-crystal X-ray diffraction analysis has confirmed their structures. Supramolecular interactions in both complexes were explored in Figure 1 [7]:

Similarly, in [15-crown-5⊃Ca^II^–(μ-OH)–Mn^III^MST]^+^ complexes, calcium ions are bound with the [MST]^3^ ligand of the Mn complex ([MST]^3−^-tripodal ligand N,N′,N″-[2,2′,2″-nitrilotris(ethane-2,1-diyl)]tris(2,4,6-trimethylbenzenesulfonamido) [8]. Rate enhancements for the reduction in dioxygen by a Mn^II^ complex were observed in the presence of redox-inactive group 2 metal ions. 

Pyridine diimine (PD) iron complexes containing 15-crown-5 that are located in the secondary coordination sphere were synthesized and characterized. Minor shifts in the reduction potential of the metal ligand framework were observed during the encapsulation of either Na^+^ or Li^+^ (Figure 2) [9]:

In [10], the authors achieved unprecedented redox switching in the two-tier Gd^III^ phthaocyanine complex, including crown ethers, which is due to the outer-sphere inclusion of potassium cations between the decks with the simultaneous twisting of the ligands. A structural change leads to an increase in the distance between the decks and dramatically facilitates the reduction in phthaocyanine ligands. Interestingly, only the inclusion of cations of potassium salts of weak acids (KOPiv and KOAc) into crown ethers causes a redox switch, in contrast to salts of strong acids (KBr, KOPic, KSCN, and KPF6), where such a redox process does not occur.

In the cases discussed above, the coordination of salts of redox-inactive metals (M′L) (M′–redox-inactive metal) with complexes of transition metals (ML^1^) is carried out through the ligand (L^1^) of a transition metal complex:

ML^1^–M′L (L^1^–M′ bonds) (I type).

Cases where the anions of salts of redox-inactive metals (M′L) are coordinated with ions of transitional metal complexes (ML^1^) (nickel- or cobalt-acetylacetonates),

L^1^M–L M′ (M–L bonds) (II type)), are known.

In the works by the authors of [11,12], kinetic and UV spectrophotometric evidence for the formation of Ni(acac)_2_ complexes with NaSt was obtained (type II), namely, (acac)_2_Ni-StNa. The formation of M-O-Mg bonds (M = Co or Cu) in analogy complexes between cobalt and copper acetylacetonates and alkaline earth metal naphthenates (magnesium) was proven in [13].

The chemical industry is often a source of dangerous environmental pollution, and in this regard, it is in this area that it is especially important to develop new environmentally friendly chemical processes characterized by a reduced level of energy consumption and the minimal formation of by-products. The development of green industrial process-selective hydrocarbon oxidation by O_2_ is determined by the ability of researchers to control these processes. An effective method of controlling the rate and mechanism of the free radical oxidation of hydrocarbons is the use of a catalyst.

The problem of the selective oxidation of alkylarenes into hydroperoxids, which are primary oxidation products, is theoretically and economically sound. Hydroperoxides are used as intermediates in the large-scale production of important monomers. For instance, propylene oxide and styrene are synthesized from α-phenyl ethyl hydroperoxide, and cumyl hydroperoxide is the precursor in the synthesis of phenol and acetone. Matienko L.I. was the first to propose a method for modifying liquid-phase metal complex catalysts by additives of mono- or multidentate electron-donating activating ligands in order to increase the selectivity and conversion of the oxidation of alkylarenes (ethylbenzene and cumene) by molecular oxygen into corresponding hydroperoxides [11,12]. The research on the mechanism of action of such modifying ligands led to the creation of effective catalysts of the selective oxidation of ethylbenzene to α-phenyl ethyl hydroperoxide [11,12]. Heteroligand nickel complexes, Ni(acac)_2_·L^2^ (L^2^ = electron-donating activating ligand), are not only efficient catalysts for the oxidation of alkylarenes to hydroperoxides but are also structural and functional models of Ni-ARD Acireductone dioxygenase.

The unusual effect of a significant increase in conversion (***C***) and an increase in selectivity (***S***) during the oxidation of ethylbenzene into α-phenyl ethyl hydroperoxide (PEH) was discovered when PhOH was added to the {Ni(acac)_2_ + L^2^} system (L^2^ = electron-donating ligand). In this case, the catalytic efficiency of triple systems {Ni(acac)_2_ + MSt + PhOH}, estimated by the values of ***C*** = 20–35% and selectivity ***S***_PEH,max_ = 85–87%, [PEH]_max_, significantly exceeded the efficiency of other triple catalytic systems, {Ni(acac)_2_ + L^2^ + PhOH} (L^2^ = NMP, HMPA, or DMF), and most active binary systems. (Ref. [12]; Patent RU 2237050, Matienko L.I., Mosolova L.A. (2004)).

In this manuscript, we obtained new kinetic patterns of ethylbenzene oxidation catalyzed by three-component {Ni(acac)_2_ + MSt + PhOH} systems (M = Li, Na, or K). An analysis of kinetic patterns makes it possible to evaluate the activity of the resulting complexes, {Ni(acac)_2_∙MSt∙PhOH}, with respect to O_2_, PEH, and RO_2_^•^ radicals. We used the AFM method to prove the possibility of the formation of stable supramolecular structures and the significance of outer-sphere regulatory interactions during the oxidation of ethylbenzene into α-phenyl ethyl hydroperoxide in the presence of triple systems, {Ni(acac)_2_∙MSt∙PhOH}. The obtained kinetic data allowed us to draw conclusions about the unique mechanism of catalysis by triple complexes including Ni; the redox-inactive metals Li, Na, and K; and PhOH.

## 2. Results

We previously found that the addition of alkali metal stearates (MSt) increased the activity of Ni(acac)_2_ as a catalyst for the oxidation of ethylbenzene in PEH [12]. Like electron-donating ligands (NMP, HMPA, DMF, crown ethers, and quaternary ammonium salts), MSt additives increase the initial rate of ethylbenzene oxidation by O_2_, catalyzed by Ni(acac)_2_. This is due to the higher activity of the resulting {Ni(acac)_2_•MSt} complexes at the stages of free radical formation (at the micro stages of the chain initiation (O_2_ activation) and radical decomposition of PEH) compared to the action of Ni(acac)_2_. This is stage I of the oxidation process.

An increase in the initial rates of oxidation in the line LiSt < NaSt < KSt may be associated with an increase in the electron-donating activity of MSt as the activating ligands in this line [12]. With this, the participation of the catalyst Ni(acac)_2_•MSt into the micro steps of chain propagation (Cat + RO_2_^•^→) (at Cat participation) and quadratic chain termination (RO_2_˙ + RO_2_˙→) cannot be excluded [12]. The results obtained in [8] illustrate the possibility that oxidation-inactive group 2 metal ions can increase the rate of O_2_ reduction by the Mn(II) complex.

The subsequent transformation of the nickel catalyst Ni(acac)_2_•L^2^ into more selective active complexes is a result of reactions in the outer-coordination-sphere nickel complexes [12].

When studying the influence of axial-modifying ligands on the activity of homogeneous metal complex catalysts, researchers usually pay attention to their steric and electronic properties. The interaction in the outer coordination sphere and the role of hydrogen bonds and other non-covalent interactions are practically not taken into account [14,15].

Thus, the above-discussed mechanism of catalysis by {Ni(acac)2-MSt} complexes describes stage I of the oxidation process (catalytic particles P_1_). The studied kinetic patterns suggest that the increase in selectivity (S_REH_) in the process of ethylbenzene oxidation, catalyzed by {Ni(acac)_2_•MSt} (up to S_REH_,_max_), may be associated with the greater activity of the products of the transformation of primary {Ni(acac)_2_•MSt} complexes (stage II of the process). The formation of the active form of the catalyst may be the result of the outer-sphere regioselective addition of O_2_ to the nucleophilic γ-C atom of one of the (acac)^−^ ligand’s {Ni(acac)_2_•MSt} complex (by analogy with the mechanism of transformation of {Ni(acac)_2_•L^2^} nickel complexes (L^2^—different mono- and multidentate electron-donating ligands)). This transformation mechanism is similar to the mechanism of action of NiARD dioxygenase [12] (see below). The structure of the resulting active selective {Ni_x_(acac)_y_(OAc)_z_MSt_n_•H_2_O_m_} complexes (an analogue of the {Ni_2_(acac)(OAc)_3_NMP•2H_2_O} complex (L^2^ = NMP)) has been proposed (catalytic particles P_2_). In stage III, the intermediate {Ni_x_(acac)_y_(OAc)_z_MSt_n_•H_2_O_m_} complexes are presumably oxidized into Ni(OAc)_2_—a product of complete Ni(acac)_2_ oxidation with dioxygen. In this case, the selectivity of ***S***_REH_ decreases, since Ni(OAc)_2_ is a catalyst of the heterolytic decomposition of PEH [12] (catalytic particles P_3_). Schematically, the transformation process of {Ni(acac)_2_-MSt} (P_1_) complexes can be represented as follows:P_1_ → P_2_ → P_3_

### 2.1. The Mechanism of Ethylbenzene Oxidation Catalyzed by {Ni(acac)_2_•MSt•PhOH} Triple Systems

This article presents new kinetic data obtained by GLC using the computer programs Mathcad and Graph2Digit for processing experimental data in studying the mechanism of ethylbenzene oxidation catalyzed by {Ni(acac)_2_ + L^2^ + PhOH} triple systems (L^2^ = LiSt or KSt).

As mentioned above, in the presence of the three-component system {Ni(acac)_2_ + L^2^ + PhOH}, a significant catalytic effect was established—a synergistic increase in the parameters degree of conversion ***C***, (***S***_PEH_)_max_, and [PEH]_max_ compared to catalysis by a binary catalytic system. The synergistic effects of increasing ***C***(from 12–14% to 20–35%), as well as (***S***_PEH_)_max_ (from 75–82% to 85–87%) upon the introduction of phenol into the {Ni(acac)_2_ + L^2^} catalytic system, indicated an unusual catalytic activity of the resulting triple complexes [12].

The analogy in the course of ethylbenzene oxidation reactions in the presence of the {Ni(acac)_2_ + MSt + PhOH} system and in the presence of the {Ni(acac)_2_∙L^2^∙PhOH} complex (L^2^ = NMP) suggests that the same mechanism of selective catalysis is realized for both reactions. Namely, catalysis is carried out by triple complexes formed during the oxidation of {Ni(acac)_2_∙L^2^∙PhOH} } [12].

The proposed structure of the heteroligand triple complex {Ni(acac)_2_·MSt·PhOH}, shown in Figure 3, is based on our previously obtained experimental (UV spectroscopy) results [12,16] and literature data [13]:

The phenomenon of the dual role of phenol coordinating with the nickel complex, depending on the ligand environment of the metal ion, is described in detail in [12,16]. Phenol can act as a deactivating agent if it coordinates with Ni(acac)_2_. {Ni(acac)_2_•PhOH} complexes are effective oxidation inhibitors. In the presence of the electron-donating extra ligand L^2^, the resulting triple complexes including phenol, {Ni(acac)_2_∙L^2^∙PhOH}, are effective catalysts for the oxidation of ethylbenzene into PEH.

Figure 4 shows the kinetics of the accumulation of (1) PEH, (2) acetophenone (AP), and (3) methylphenylcarbinol (MPC) during ethylbenzene oxidation catalyzed by the (a) {Ni(acac)_2_ + NaSt + PhOH} and (b) {Ni(acac)_2_ + LiSt + PhOH} triple systems.

An analogy in the course of the oxidation reactions of ethylbenzene in the presence of the {Ni(acac)_2_ + L^2^ + PhOH} systems (L^2^ = NMP, HMPA, or DMF) was observed: the constancy of the reaction rate of the ethylbenzene oxidation catalyzed by the {Ni(acac)_2_ + MSt + PhOH} triple system for a long time. The rates of formation of PEH, AP, and MPC are constant up to t > 30 h (MSt = NaSt) and t ≥ 30 h (MSt = LiSt). The extreme stability of the resulting {Ni(acac)_2_∙MSt∙PhOH} triple complexes is apparently due to the (acac) ligand in the triple complexes (Figure 4), which is stable and does not undergo transformation for a long time [12,16].

We studied the mechanism of the formation of the products of ethylbenzene oxidation catalyzed by {Ni(acac)_2_•MSt•PhOH} complexes (M = Na, Li, or K) using the method of graphical differentiation. An assessment showed that the ratio of the rates of accumulation of the AP (or MPC) by-products and the main product PEH at t → 0 is non-zero, *w*_AP(MPC)_/*w*_PEH_ ≠ 0 at t → 0, i.e., all three products AP, MPC, and PEH are formed in parallel (as an example, see Figure 5a,c). Similar regularities are typical for catalysis by {Ni(acac)_2_•L^2^•PhOH} triple systems (L^2^ = HMPA, NMP, or DMF) [12]. We also observed a parallel formation of AP and MPC (*w*_AP_/*w*_MPC_ ≠ 0 at t → 0) in the case of catalysis by {Ni(acac)_2_•LiSt(KSt) •PhOH} systems (for example, see Figure 5b). However, in the case of catalysis by the {Ni(acac)_2_•NaSt•PhOH} complexes, *w*_AP_/*w*_MPC_ → 0 at t → 0, i.e., a sequential formation of AP from MPC was observed [16].

Thus, AP and MPC in ethylbenzene oxidation catalyzed by {Ni(acac)_2_•MSt•PhOH} are not products of PEH decay. AP and MPC apparently form in parallel with PEH, into the reactions of chain propagation (Cat + RO_2_^•^→) (at Cat participation) and quadratic chain termination (RO_2_˙ + RO_2_˙ →).

We propose, as one probable form, a “latent-radical” mechanism for the formation of products as result the chain propagation reaction (Cat + RO_2_˙→) (by analogy with the works [12,16]).

A “latent-radical” mechanism of reaction (Cat + RO_2_˙→) (Cat = M) includes the formation and hemolytic decomposition of the intermediate {ROO-M} complex through the O–O bond [17,18]. Figure 1 shows the {ROO-M} complex decomposition. The radicals RO• and MO•, formed during {ROO-M} complex decomposition in the cell of the solvent, either recombine in the cell (R’ = O (AP) and HOM are formed) or leave the cell to form ROH (MPC) and reduce M = Cat (the last two reactions).

Thus, we have shown that the {Ni(acac)_2_∙MSt∙PhOH} triple complexes, unlike binary Ni(acac)_2_∙MSt complexes, were not active in the reaction with hydroperoxide but were active in the reactions of chain initiation (O_2_ activation) and chain propagation (Cat + RO_2_**^•^**→) (and, probably, in chain termination). In these systems, O_2_ activation may also be promoted through the formation of intramolecular H bonds. We established the role of intramolecular H bonds in the mechanism of the formation of triple catalytic {Ni(acac)_2_∙L^2^∙PhOH} complexes (L^2^ = NMP) in ethylbenzene oxidation by O_2_ [12].

### 2.2. The Intermolecular Role of H Bonds and Supramolecular Structures in the Mechanism of Ethylbenzene Oxidation Catalyzed by {Ni(acac)_2_∙L^2^∙PhOH} Systems

A characteristic feature of {Ni(acac)_2_∙MSt∙PhOH} triple complexes as catalysts is their high efficiency due to the absence of oxidative transformations of catalysts during the reaction. The coordination of phenol with {Ni(acac)_2_∙MSt} complexes apparently leads to the formation of triple complexes in which the reaction of O_2_ incorporation into the acac-ligand and the subsequent oxidative transformation of the nickel complex becomes unlikely. The high stability of the triple complexes may also be due to the formation of intermolecular hydrogen bonds (phenol carboxylate) and, possibly, other non-covalent interactions that facilitate the formation of stable supramolecular structures.

Supramolecular chemistry is a field of research that has seen significant developments in recent years. A concept has been developed for studying intermolecular bonds, including coordination, and halogen and hydrogen bonds at supramolecular self-organization on the surface [19,20,21,22]. Studies of cytochrome P450-dependent monooxygenases, which are part of the class of hemoproteins, have shown that the coordination spheres of the active centers of proteins play a decisive role in determining the properties of the metal cofactor [21,22,23].

The mechanism of catalysis often involves the formation of supramolecular structures during the catalytic reaction. It has been suggested (L.I. Matienko) that the high efficiency and stability of the active forms of catalysts in ethylbenzene oxidations may be associated with the formation of stable supramolecular structures due to intermolecular H bonds and, possibly, other non-covalent interactions. A new approach was proposed by us to assess the role of H bonds and supramolecular structures in the mechanisms of homogeneous and enzymatic catalysis—the AFM method [21,22,23].

So, it was possible to assume that the long-term activity of Ni(acac)_2_∙MSt∙PhOH is likely due to the formation of stable supramolecular structures as a result of intermolecular hydrogen bonds and possibly other non-covalent interactions. It is known, for example, that the L-leucine-derived ligand (H_2_L(L-leu)), KOH, and Ni^II^ salt in a 2:2:1 ratio self-assembled into a rather large (approximately 13 Å) supramolecular assembly with the formula [K{Ni(HL(L-leu))_2_}_3_ ]^+^ (1). Visible spectroscopic studies showed a retention of assembly (1) in DMF, which is stable even after the addition of 5 equivalents of 18-crown-6 [24,25].

The possibility of the association of {Ni(acac)_2_∙MSt∙PhOH} triple complexes to supramolecular structures due to H bonding follows from an analysis of AFM data (Figure 6 and Figure 7). The formation of supramolecular structures on the surface is due to the balance between the interactions of molecules with the surface and intermolecular interactions, which may be the result of hydrogen bonds and other non-covalent interactions. In the course of scanning investigated samples, it was found that the structures are fixed on the surface strongly enough due to H bonding. The self-assembly-driven growth of the supramolecular structures, based on Ni(acac)_2_∙NaSt(or LiSt)∙PhOH complexes due to H bonds and perhaps other non-covalent interactions, was observed on the silicone surface. It does not follow, from the data obtained, that exactly such structures are formed under real conditions of catalytic oxidation. But the self-organization of supramolecular structures based on catalytically active heteroligand complexes of nickel {Ni(acac)_2_∙MSt∙PhOH} on the surface apparently indicates a high probability of the formation of such structures in the real conditions of catalytic oxidation.

Thus, the high efficiency of {Ni(acac)_2_ + MSt + PhOH} systems as selective catalysts for the oxidation of ethylbenzene in PEH, as one reason, may be associated with the self-organization of the resulting Ni(acac)_2_∙MSt∙PhOH complexes into extremely stable supramolecular structures due to intermolecular H bonds (phenol-carboxylate) and possibly other non-covalent interactions. The higher effectivity of Ni(acac)_2_∙NaSt∙PhOH complexes as selective catalysts, as compared with Ni(acac)_2_∙LiSt∙PhOH, seems due to the formation of more stable supramolecular structures in the first case. The data in Figure 6 and Figure 7 testify in favor of this.

As mentioned before, we have established that the transformation of Ni(acac)_2_∙L^2^ complexes into more active catalytic particles in the process of ethylbenzene oxidation by O_2_ occurs by a mechanism similar to the action of NiARD [12].

By the AFM method (Figure 7b), the formation of stable supramolecular structures was recorded based on model Ni(acac)_2_∙L^2^∙L^3^, L^2^ = His; L^3^ = Tyr) complexes containing the amino acids L-histidine (L^2^ = His) and L-tyrosine (L^3^ = Tyr). UV spectroscopy data indicate the intra- and outer-sphere coordination of the extra-ligands L^2^ and L^3^ with M(acac)_n_ (M = Ni or Fe; *n* = 2,3; L^2^ = NMP or His; L^3^ = Tyr) [16,21]. The formation of supramolecular structures seems due to the formation H bonds and possibly other non-covalent interactions. This may indicate the possible role of the Tyr fragments in the stabilization of primary NiARD complexes as one of the possible regulatory factors in the mechanism of Ni(Fe)ARD action [22,23,26,27,28].

## 3. Discussion

The mechanism of catalysis by the heteroligand complexes (A) {Ni(acac)_2_∙L^2^} and (B) {Ni(acac)_2_∙L^2^∙PhOH} (L^2^ = MSt (M = Li, Na, or K)) in the processes of the selective oxidation of ethylbenzene by molecular oxygen was studied by kinetic methods. This is due to the detected effect of increasing the oxidation-reduction activity of Ni(acac)_2_ in the processes of the selective oxidation of ethylbenzene to PEH in the presence of MSt additives, including redox-inactive metals (M = Na, Li, or K). An unusual effect of a significant increase in the conversion of ethylbenzene oxidation to PEH, while maintaining ***S***_PEHmax_ at a level of 85–87%, was also detected when introducing the third component of phenol into the {Ni(acac)_2_ + L^2^} system. The formation of A and B complexes was proven kinetically and by UV spectroscopy. It was interesting to compare the activity of A and B complexes towards O_2_, ROOH, and RO_2_^•^ radicals. We have established that the {Ni(acac)_2_∙MSt∙PhOH} triple complexes, unlike the {Ni(acac)_2_∙MSt} (A) binary complexes, are not active in the reaction with ROOH but are active in chain initiation (O_2_ activation) and chain propagation (Cat + RO_2_^•^→) and chain termination reactions. In these systems, O_2_ activation can be promoted by the formation of intramolecular hydrogen bonds. More recently, “it has been recognized that further control over activity and selectivity can be achieved by using the “second coordination sphere”, which can be seen as the region beyond the direct coordination sphere of the metal center. Hydrogen bonds appear to be very useful interactions in this context as they typically have sufficient strength and directionality to exert control of the second coordination sphere, yet hydrogen bonds are typically very dynamic, allowing fast turnover” [15]. The advantage of the {Ni(acac)_2_ + L^2^ + PhOH} triple systems (L^2^ = MSt or other electron-donating ligands) is that the triple complexes formed in situ are active for a long time, while the acac–ligand is stable and does not undergo transformation during ethylbenzene oxidation. The higher effectivity of {Ni(acac)_2_∙MSt∙PhOH} complexes as selective catalysts of ethylbenzene oxidation into PEH, as one reason, may be due to the self-organization of triple complexes in the course of ethylbenzene oxidation into extremely stable supramolecular structures due to intermolecular H bonds (phenol–carboxylate) and, possibly, due to other non-covalent bindings. The assumed significance of outer-sphere interactions and the self-organization of B complexes into stable supramolecular structures was confirmed by the AFM method. The self-organization of supramolecular structures based on catalytically active heteroligand complexes of nickel {Ni(acac)_2_∙MSt∙PhOH} on the surface (AFM) apparently indicates a high probability of the formation of similar structures in the real conditions of catalytic oxidation.

The catalytic active A and B complexes are also structural and functional models of the active site of NiARD dioxygenase. Ni(Fe)-ARD enzymes participate in the methionine salvage pathway (MSP), a universal pathway for converting sulfur-containing metabolites into methionine. ARD exhibits structural and functional differences depending upon the metal bound in the active site. In the penultimate step of the MSP, the Fe^2+^-bound form of ARD catalyzes the on-pathway oxidation of acireductone, leading to methionine, whereas Ni^2+^-bound ARD catalyzes an off-pathway reaction, producing methylthiopropionate and carbon monoxide, a biological signaling molecule and an anti-apoptotic [26]. We suggest that the role of the second coordination sphere, which includes the Tyr fragment, should be taken into account in the case of the functioning of Ni(Fe)ARD. With the AFM method, convincing evidence was obtained at the model level in favor of the participation of the tyrosine fragment (in the outer sphere of the ARD enzyme) and supramolecular structures as regulatory factors in the mechanism of action of Ni(Fe)ARD.

In addition to its enzymatic function, several studies have indicated the role of ARD enzymes in carcinogenesis and tumor metastasis. In addition, several biochemical and genetic studies have shown that human ARD suppresses cancer development [28].

Recently, the high antitumor activity of the mixed-ligand nickel complex [Ni(L^1^)_2_L^2^]H_2_O (L^1^ is the acetylacetonate ion (acac), and L^2^ is 2-aminopyridine) in vitro against the gastric cancer cell line MKN45 was established [29].

## 4. Materials and Methods

Ethylbenzene (RH = C_6_H_5_CH_2_CH_3_) was oxidized by O_2_ at 120 °C in a glass bubbling-type reactor in the presence of catalytic systems. An analysis of the oxidation products included the following. *α*-Phenyl ethyl hydroperoxide (PEH = C_6_H_5_HCOOH(CH_3_)) was analyzed by iodometry. The by-products methylphenylcarbinol (MPC = C_6_H_5_HCOH(CH_3_)), acetophenone (AP = C_6_H_5_CO(CH_3_)), phenol (PhOH = C_6_H_5_OH), and also RH during oxidation reaction were examined by GLC (Russia) [11].

Experimental data processing was performed using the special computer programs Mathcad and Graph2Digit. The order of PEH, AP, and MPC formation was determined from the time dependence of the ratios of product accumulation rates at t → 0. The variation of these ratios with time was evaluated by graphical differentiation from the experimental dependences Δ[P_1_]_ij_/Δ[P_2_]_ij_ from t_j_. Δ[P_1_]_ij_ and Δ[P_2_]_ij_ are the increase in P_1_ and P_2_ concentrations for Δt = t_j_ – t_i_.

In the AFM study, the scanning probe microscope SOLVER P47 SMENA10 (Administrative District Zelenograd of Moscow, 124490, Russia) was used, using an NSG30 cantilever with a radius of curvature of 10 nm, a tip height of 10–15 µm, and a cone angle of ≤22° in taping mode, and a resonant frequency of 150 KHz was used.

We used an NSG30_SS cantilever (Nanosensors^TM^ Advanced Tec^TM^ AFM probes, Neuchatel, Switzerland) with a radius of curvature of 2 nm, a resonance frequency of 300 kHz, and a force constant of 22–100 N/m. TipsNano, in tapping mode, used for the AFM research of supramolecular structures.

As the substrate, a chemically modified polished silicone surface was used.

The waterproof, modified silicone surface was exploited for the self-assembly-driven growth due to the H bonding of Ni(acac)_2_∙NaSt(or LiSt)∙PhOH complexes with the silicone surface. The saturated chloroform (CHCl_3_) solution of the Ni(acac)_2_∙NaSt(or LiSt)∙PhOH complex (1:1:1) was put on a surface and maintained for some time, and then, the solvent was deleted from the surface by means of a special method—a spin-coating process.

Due to the poor solubility of potassium stearate in chloroform (as well as in other organic solvents), it was not possible to investigate the possibility of the self-organization of the Ni(acac)_2_•KSt•PhOH complex into supramolecular structures on the surface.

## 5. Conclusions

The advantage of {Ni(acac)_2_ + L^2^ + PhOH} triple systems (L^2^ = MSt or other electron-donating ligands) is that the {Ni(acac)_2_∙MSt∙PhOH} triple complexes formed in situ are active for a long time, and the acac–ligand is stable and does not undergo transformation during ethylbenzene oxidation. We hypothesized that the high efficiency and stability of Ni(acac)_2_∙MSt∙PhOH (M = Li or Na) complexes as selective catalysts for the oxidation of ethylbenzene in PEH may be related, as one reason, to the formation of stable supramolecular structures due to intermolecular H bonds and possibly other non-covalent interactions. Indeed, the observed self-organization of supramolecular structures based on {Ni(acac)_2_∙MSt∙PhOH} complexes on the surface (AFM method) seems to indicate a high probability of the formation of similar structures under real conditions of catalytic oxidation.

The study of the mechanism of catalysis of ethylbenzene oxidation by heteroligand ternary nickel complexes, Ni(acac)_2_∙MSt∙PhOH (M = Li, Na, or K), has shown that these complexes are not only an effective catalyst for the selective oxidation of ethylbenzene to PEH. In addition, these complexes are structural and functional models of the active site of NiARD dioxygenase. The results of AFM studies may also be useful to explain the mechanism of action of Ni(Fe)-ARD enzymes. One of the possible ways to regulate the activity of NiARD enzymes may be the formation of outer-sphere H bonds and multidimensional forms at the expense of H bonds.

## Data Availability

All of the experimental data presented belong to the authors of this manuscript and are available.

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
