# Peer review of "Selective Catalysis by Complexes Including Ni and Redox-Inactive Alkali Metals (Li, Na, or K) in Oxidation Processes: The Role of Hydrogen Bonds and Supramolecular Structures"

_ijms, 2025, doi:10.3390/ijms26031166_

Round 1
Reviewer 1 Report
Comments and Suggestions for Authors
This paper describes the mechanisms of oxidation of ethylbenzene catalyzed by nickel(II) complexes containing redox-inactive alkali-metal ions. This is a carefully done study and the findings are interesting. Thus, this paper is worth publishing in International Journal of Molecular Science with minor revisions. Some additional comments are listed below.
1) page 1, line 21: ttiple --> triple
2) page 1, line 41-43: O2− --> O2− or O2−
3) page 2, line 64-65: Al(3+) --> Al3+
4) page 3, line 115: The word “ahhfrently” is misspelled.
5) page 4, line 140 and 151: oxidation to the to a-phenyl --> oxidation to a-phenyl
6) page 8, line 277-278: This line should be italic because this is the title of section 2-2.
7) page 9, line 311: H(2)L(L-leu) --> H2L(L-leu)
8) page 9, line 313: [K{Ni(HL(L-leu))(2)}(3)](+)(1) --> [K{Ni(HL(L-leu))2}3]+ (1)
Author Response
Комментарий 1:
- страница 1, строка 21: ttiple --> тройной
Ответ на комментарий 1:
Ошибка исправлена.
Комментарий 2:
- стр. 1, строка 41-43: O2−--> O2− или O2−
Ответ на комментарий 2:
Такие ошибки не наша вина. Тем не менее, мы исправили текст. Часть текста была удалена, поскольку он содержит ненужные детали, не относящиеся к содержанию статьи.
Комментарий 3:
- страница 2, строка 64-65: Al(3+) --> Al3+
Ответ на комментарий 3:
Я написал это так, как написали авторы. Но я исправил с учетом мнения рецензента 2, который настаивал на том, что написание цифр должно быть одинаковым для всех металлов в этой ссылке: MnIII-(μ-O)2-Mn IV и AlIII. Страница 2, строки 12-20.Комментарий 4:
- страница 3, строка 115: Слово "ahhfrently" написано с ошибкой.
Ответ на комментарий 4:
Спасибо за наблюдение. Мы убрали это слово из текста.
Комментарий 5:
- Страница 4, строки 140 и 151: Окисление до А-фенила --> Окисление до А-фенила
Ответ на комментарий 5:
Ошибка исправлена.
Комментарий 6:
- страница 8, строка 277-278: Эта строка должна быть выделена курсивом, потому что это название раздела 2-2.
Ответ на комментарий 6:
Эта ошибка не моя вина. Но я исправил ошибку.
Комментарий 7:
- страница 9, строка 311: H(2)L(L-leu) --> H2L(L-лей)
Ответ на комментарий 7:
Оригинальное написание совпадает с авторским. Но я изменил его согласно вашему комментарию
Комментарий 8:
8) страница 9, строка 313: [K{Ni(HL(L-leu))(2)}(3)](+)(1) --> [K{Ni(HL(L-leu))2}3]+ (1)
Ответ на комментарий 8:
Я изменил его, как вы предложили.

Reviewer 2 Report
Comments and Suggestions for Authors
The manuscript “ Selective Catalysis by Complexes Including Ni and Redox-Inactive Alkali Metal (Li, Na, or K) in Oxidation Processes. The Role of Hydrogen Bonds and Supramolecular Structures” is interesting. However, before considering publishing, several corrections must be performed.
1. The manuscript appears as two big parts without cohesion. The first part corresponds to the introduction, the other is the results and discussion. A lot of examples of reactions and catalysts are given in the introduction that do not have a direct relation to the results. If the authors insist on the importance of the examples for the results, an improvement of the redaction must be made.
2. The redaction is very difficult to read; several English grammar and spelling were located.
3. Page 1, line 10: no correspondence author is mentioned.
4. Several chemical mistakes were located; for example, on page 1, lines 41 and 42, it is written O2-. It is not clear if the 2 must be a subscript or a superscript; I mean whether the 2 indicates a di-molecular oxygen anion of charge minus one or an oxygen atom with charge minus two.
5. Other chemical formulae also have errors of this type.
6. Authors must homogenize the form of writing the oxidation state, of particular of metals ions. Some oxidation states are written in Roman numerals (upper case or lower case) or in normal numbers. For example, manganese (II), manganese (iv), or Al (3+) on page 2, lines 64 to 67.
7. On Page 2, line 84, the small rabbit is very nice, but it should be depicted as a figure; add a footnote and a description (including the name, as in Fig. 2).
8. The same goes for the structures of page 3, line 97.
9. Several text sections are highlighted in light gray; remove the highlighting.
10. Page 4, lines 135 to 138, why is the text in bold type?
11. Results and discussion are poor; the text is full of reported examples, but no proper demonstration on the paper is made and supported experimentally.
12. Stages I and II are mentioned in the text, but it is unclear what they are. An image or scheme must be included.
13. The quality of Fig. 1 is poor. It gets lost in the text what is [C] and their units. These indications must be evident in the axes of Fig. 1.
14. Modify Fig. 1 to present the bullets and lines. As it is now (just lines), it is unclear whether there is an adequate amount of experimental points. I suspect there are very low amounts of experimental points in the whole time range.
15. Figure 3 is poorly explained and discussed.
16. Scheme 1 is of low quality; reactions must be numbered. Reactions must be explained in the text.
17. Figure 4. I am not convinced why a micrometer image might provide correct information about complexes in the size range of 13 Amstrong. By the way, the unit A is incorrect (Å, page 9. Line 312). A proper justification must be made; otherwise, the experiment result is nonsense.
18. Conclusion are not supported by the experimental results, they are based on results from reported works.
19. Page 11, lines 400-409. This paragraph is more suitable for discussion rather than an experimental section.
Comments on the Quality of English LanguageThe English is very poor. Authors, please consider a guide with a professional editor.
Author Response
Comment 1:
- The manuscript appears as two big parts without cohesion. The first part corresponds to the introduction, the other is the results and discussion. A lot of examples of reactions and catalysts are given in the introduction that do not have a direct relation to the results. If the authors insist on the importance of the examples for the results, an improvement of the redaction must be made.
Response to the comment 1:
We have restructured the paper to meet the requirements of the editors. In the introduction, we show what is known in the literature about studies on the influence of redox inactive metals on the properties of compounds of redox active metals. These studies are related to the existence in nature of enzymes that include redox inactive metals along with transition metals. We also give a number of examples of the influence of redox-active metals on the properties of complexes of metals of variable valence, which are used as catalysts. And we turn to our catalytic triple systems, which have unusual properties associated with the couplings of the redox inactive metals and PhOH within the triple complexes. These properties are high stability and associated high conversion while maintaining high selectivity of ethylbenzene oxidation to the target product - hydroperoxide.
Comment 2:
- The redaction is very difficult to read; several English grammar and spelling were located.
Response to the comment 2:
We have considered this comment and tried to improve the English language in this article
Comment 3:
- Page 1, line 10: no correspondence author is mentioned.
Response to the comment 3:
Error corrected.
Comment 4:
- Several chemical mistakes were located; for example, on page 1, lines 41 and 42, it is written O2-. It is not clear if the 2 must be a subscript or a superscript; I mean whether the 2 indicates a di-molecular oxygen anion of charge minus one or an oxygen atom with charge minus two.
Response to the comment 4:
Such errors are not our fault. However, we have corrected the text. Part of the text has been deleted because it contains unnecessary details not relevant to the content of the article.
Comment 5:
- Other chemical formulae also have errors of this type.
Response to the comment 5:
We have corrected the text
Comment 6:
- Authors must homogenize the form of writing the oxidation state, of particular of metals ions. Some oxidation states are written in Roman numerals (upper case or lower case) or in normal numbers. For example, manganese (II), manganese (iv), or Al (3+) on page 2, lines 64 to 67.
Response to the comment 6:
I wrote it the way the authors wrote it. But I corrected taking into account the opinion of reviewer 2, who insisted that the spelling of the numbers (валентность, заряд) should be the same for all metals in this reference: Page 2, Lines 12-20.
Comment 7:
- On Page 2, line 84, the small rabbit is very nice, but it should be depicted as a figure; add a footnote and a description (including the name, as in Fig. 2).
Response to the comment 7:
I took your advice and made a caption for the figure. It's now figure 1
Comment 8:
- The same goes for the structures of page 3, line 97.
Response to the comment 8:
I took your advice and made a caption for the figure. It's now figure 8.
Comment 9:
- Several text sections are highlighted in light gray; remove the highlighting.
Response to the comment 9:
I took out the gray
Comment 10:
- Page 4, lines 135 to 138, why is the text in bold type?
Response to the comment 10:
Apparently the error occurred when reformatting the text. I've corrected the text.
Comment 11:
- Results and discussion are poor; the text is full of reported examples, but no proper demonstration on the paper is made and supported experimentally.
Response to the comment 11:
At the request of the editors, I have changed the structure of the paper. Now there are separate Results and Discussion.
Comment 12:
Stages I and II are mentioned in the text, but it is unclear what they are. An image or scheme must be included.
Response to the comment 12:
I have given a more detailed description of the mechanism of oxidative transformation of complexes in the process of ethylbenzene oxidation (last paragraph on p. 4 and the first paragraph on page 5).
Comment 13:
The quality of Fig. 1 is poor. It gets lost in the text what is [C] and their units. These indications must be evident in the axes of Fig. 1.
Response to the comment 13:
A detailed description of Fig. 1 (now Fig. 4) is given in the figure caption.
Comment 14:
- Modify Fig. 1 to present the bullets and lines. As it is now (just lines), it is unclear whether there is an adequate amount of experimental points. I suspect there are very low amounts of experimental points in the whole time range.
Response to the comment 14:
Thank you for your careful reading of the article. We used the computer programs Mathcad and Graph2Digit for processing experimental data in our last works [Ref.16]. The computational representation, in our opinion, gives a clearer picture of the development of the oxidation process. Kinetic curves with experimental points were in our earlier works (Refs. 11, 12).
Comment 15:
- Figure 3 is poorly explained and discussed.
Response to the comment 15:
Fig. 3 (now Fig. 6) presents data from calculations performed in Excel based on experimental points. The method for evaluating the mechanism of oxidation product formation is described in detail in Methods and Materials.
Comment 16:
- Scheme 1 is of low quality; reactions must be numbered. Reactions must be explained in the text.
Response to the comment 16:
Thank you very much for the comment. The presented scheme 1 is standard for describing such complex radical-chain reactions. In view of your remark, I have provided a detailed description of the scheme in the text of the article.
Comment 17:
- Figure 4. I am not convinced why a micrometer image might provide correct information about complexes in the size range of 13 Amstrong. By the way, the unit A is incorrect (Å, page 9. Line 312). A proper justification must be made; otherwise, the experiment result is nonsense.
Response to the comment 17:
Unfortunately, there was a misunderstanding. So I have changed the text in the article for clarity. What you write about 13 angstroms has nothing to do with us. This is literature data [ref. 24]. I gave a detailed description of our use of the AFM method to prove the possibility of formation of supramolecular structures due to outer-sphere H-bonds and other noncovalent interactions on pg. 10, 3rd paragraph.
Comment 18
- Conclusion are not supported by the experimental results, they are based on results from reported works.
Response to the comment 18:
I have supplemented the conclusion with our experimental data reported in this paper
Comment 19
- Page 11, lines 400-409. This paragraph is more suitable for discussion rather than an experimental section.
Response to the comment 19:
Thank you for the observation. I moved this paragraph from the experimental part to the text of the article.
Comments on the Quality of English Language
The English is very poor. Authors, please consider a guide with a professional editor.
Response to the comment:
I did my best to correct the English text

Round 2
Reviewer 2 Report
Comments and Suggestions for Authors
Authors of the manuscript “Selective Catalysis by Complexes Including Ni and Redox-Inactive Alkali Metal (Li, Na, or K) in Oxidation Processes. The Role of Hydrogen Bonds and Supramolecular Structures” have significantly improved the quality of the work. The issues raised during the revision 1 round have been explained or improved. Now, it can be published in the International Journal of Molecular Sciences. As a general recommendation, an extra revision of the English during the edition is suggested.
Comments on the Quality of English LanguageAs a general recommendation, an extra revision of the English during the edition is suggested.